# Alkali Attack on Cation-Exchange Membranes with Polyvinyl Chloride Backing and Binder: Comparison with Anion-Exchange Membranes

**DOI:** 10.3390/membranes10090228

**Published:** 2020-09-11

**Authors:** Shoichi Doi, Nobuya Takumi, Yuriko Kakihana, Mitsuru Higa

**Affiliations:** 1Astom Corporation, 1-1 Mikagecho, Syunan, Yamaguchi 745-8648, Japan; s.doi@astom-corp.jp; 2Graduate School of Sciences and Technology for Innovation, Yamaguchi University, 2-16-1 Tokiwadai, Ube, Yamaguchi 755-8611, Japan; a064vfu@yamaguchi-u.ac.jp (N.T.); kakihana@yamaguchi-u.ac.jp (Y.K.); 3Blue Energy Center for SGE Technology (BEST), 2-16-1 Tokiwadai, Ube, Yamaguchi 755-8611, Japan

**Keywords:** cation-exchange membrane, anion-exchange membrane, degradation, alkaline, polyvinyl chloride, cleaning in place, dehydrochlorination

## Abstract

Systematic alkali immersion tests of cation-exchange membranes (CEM) with polyvinyl chloride (PVC) as their backing and binder were conducted to compare that of an Anion-exchange membrane (AEM) with the same PVC materials to investigate the mechanism of dehydrochlorination. In the immersion tests, originally colorless and transparent AEM turned violet, and chemical structure analysis showed that polyene was produced by the dehydrochlorination reaction. However, the CEM did not change in color, chemical structure or membrane properties during the test with less than 1M alkali solutions. According to the Donnan equilibrium theory and the experiments using CEM and AEM, the hydroxide ion concentration in the CEM was much lower than that in the AEM under the same conditions. However, when the alkali immersion test was performed using the CEM under more severe conditions (6 M for 168 h at 40 °C), there was a slight change in the color and chemical structure of the CEM, clearly indicating that not only AEMs, but also CEMs with PVC matrixes were deteriorated by alkali, depending on the conditions.

## 1. Introduction

The electrodialysis (ED) processes using hydrocarbon ion-exchange membranes (IEMs) have been used for almost half a century to produce salts from seawater, desalination from brackish water or food and beverages, and valuable salt recovery from wastewater treatment in the chemical industry. It has also been widely applied to various types of industries [1]. While it can be considered as a mature technology, the research and development of the ED process has still been active. For example, ED applications that use monovalent ion selective membranes, such as desalting from brackish water [2], recovery of sodium formate from waste water [3], and zinc recovery from wastewater using ED enhanced with chelating agents [4], have been reported. In addition, research on reverse electrodialysis, in which salinity gradient energy is converted to electricity using IEMs, has been actively conducted [5,6,7,8,9].

In industrial ED processes, alkaline cleaning processes, such as cleaning in place, are frequently employed to control anion-exchange membrane (AEM) contamination to ensure stable and reliable operation over long periods of time. One of the authors who works for the ASTOM corporation observed a returned membrane from a commercial plant, as shown in Figure 1. They were Neosepta^®^ AMX (AEM) and CMX (CEM), which were used in ED desalination of leachate for eight-years. Original AMX was colorless and transparent and CMX was dark yellow. However, after eight-years operation, the color of the upper side of the returned AMX irreversibly changed to dark violet, but there was no color change in the returned CMX. We would like to clarify the difference in the alkali deterioration between the AEM and the CEM.

Grande et al. focused on the degradation of IEMs with respect to the commercial food and beverage industries and compared the degradation behavior of homogeneous and heterogeneous AEMs and CEMs [10,11,12,13,14,15]. The degradation of AEMs was deeply discussed and was based on the electrochemical and mechanical properties and the analytical data using Scanning Electron Microscope (SEM), Energy-Dispersive X-ray Spectroscopy (EDX) and Fourier Transform Infrared Spectrometer (FTIR). There have been some reports on the degradation of CEMs: the CEM operating in an organic acid commercial plant was analyzed in the same manner as the AEM, and a decrease in tensile strength and an increase in electrical resistance were observed. The ion-exchange groups were lost and the surface was hydrophobized, but the reason for the phenomena has not been clarified [10,11,13]. They carried out AEM and CEM immersion tests in a sodium hypochlorite solution. While quaternary ammonium groups were decomposed in AEMs, the sulfonic acid groups were not attacked in CEMs. It has also been reported that the Styrene-Di-vinyl benzene (St-DVB) crosslinked structure was eroded and became porous in both AEMs and CEMs [14].

In general, some homogeneous cation-exchange membranes (CEMs) and AEMs have been manufactured by the paste method using polyvinyl chloride (PVC) cloth as the backing [16,17]. Vasquez et al. subjected the AEM and CEM, containing PVC as the backing and binder, to 2 M NaOH immersion tests and a 0.1 M NaOH/0.1 M HCl cycle test. They showed that the quaternary ammonium salt in the AEM membrane acts as a catalyst, and dehydrochlorination of PVC with hydroxide ions occurs, which results in the formation of a polyene sequence. No change in the stress–strain curve, obtained during tensile tests, was confirmed for the CEM, and other physical properties evaluations and chemical structure analyzes of the CEM were not performed [15].

To the author’s knowledge, for the CEMs, there have been no reports of systematic discussion of the correlation between alkaline immersion conditions and changes in the chemical structure and properties after the immersion test.

This study aims to investigate the influence of alkaline attack on the performance of commercial CEMs containing PVC as the backing and binder, and to gain some insight into the alkali degradation mechanism of CEMs and AEMs with PVC materials. To this end, the hydroxyl ion distributions inside the AEM and CEM after immersion in various concentrations of alkali solutions were measured. Subsequently, alkaline immersion tests of the CEMs were systematically conducted using the same procedure as reported for AEMs previously [18]. Further, the precursors of CEMs and AEMs were also immersed in alkali for a control. The chemical structure analyses (FTIR and X-ray Fluorescence (XRF)) and membrane properties of CEMs were also examined in order to compare them with those of AEMs.

## 2. Materials and Methods

### 2.1. Sample Membrane

Neosepta^®^ CMX was used as the commercial homogeneous CEM, and Neosepta^®^ AMX was used as the homogeneous AEM (both supplied by ASTOM Corporation of Tokyo, Japan) in this study. CMX and AMX are both standard grades for industrial use, and their basic properties are listed in Table 1. The precursors of CEM and AEM were also supplied by ASTOM Corporation (Tokyo, Japan). Both CMX and AMX were produced by the paste method using the respective precursors and contain PVC as the backing and binder [16,17].

### 2.2. Estimation of Hydroxyl Ion Concentration in CMX and AMX

All the reagents were obtained from FUJIFILM Wako Pure Chemical Corporation (Osaka, Japan) and used without further purification. The electrical conductivity of the de-ionized water used in this study was approximately 5 μS/cm. CMX and AMX were immersed in 0.01 M, 0.1 M, 1 M and 6 M NaOH solutions, and the hydroxyl ion concentrations in the membranes were determined experimentally. The 6 M NaOH concentration was almost the same as those for the fixed ionic groups of the two membranes. Firstly, the test pieces of the membranes were soaked in a 0.5 M NaCl solution to exchange the counter ion of CMX with Na^+^ ions and that of AMX with Cl^−^ ions completely. The test pieces were then immersed in 0.01 M, 0.1 M, 1 M, and 6 M NaOH solutions for 1 h at 25 °C. They were removed from the solution, rinsed with deionized water, and the surface water on the membranes was removed with a wiper. Each sample was then immersed in a 0.5 M NaCl solution with a volume of *V_o_* for 30 min. After retrieving each test piece, the change in pH in the solution was measured using a pH meter D-51 (HORIBA, Kyoto, Japan) to calculate the hydroxyl ion concentration in the solution *C_s_*. The test pieces were weighed at a wet state and were dried at 60 °C using a vacuum oven for over 24 h to measure the dry weight. The water content of the sample *V_m_* was calculated by subtracting the dry weight from the wet weight of the test piece. The OH (hydroxyl ion) concentration in the test piece *C_m_* was calculated using the following equation:(1)$$C_{m}=\;\frac{C_{s}V_{o}}{V_{m}}$$

### 2.3. Alkaline Immersion Test

#### 2.3.1. Low OH Concentration Test

An alkaline immersion test of CMX was conducted using the same methods as those used for AMX, which was described in previous papers [18,19]. CMX and AMX are recommended to be used at a temperature of under 40 °C, according to the supplier’s instructions. However, in a previous study, besides conducting the experiment at the recommended temperature of 40 °C, the authors also conducted an immersion test of AMX at 60 °C and 80 °C as an accelerated test. It was observed during the test that the water content of AMX increased, and its electrical resistance decreased, even when it was immersed in alkali-free water. It will be due to the fact that the effects of the residual stress in the cross-linked structure formed during the production being relieved by the immersion tests at high temperatures were superimposed [18]. Therefore, in this study, a CMX immersion test was performed at only 40 °C to obtain the test pieces for subsequent chemical analysis. The alkali concentrations of the low OH concentration test were 0.01 M, 0.1 M, and 1 M NaOH. A test was also performed using deionized water to serve as a control. The immersing time periods were 3 h, 24 h (1 day), and 168 h (1 week).

#### 2.3.2. High OH Concentration Test

According to the Donnan exclusion, the concentration of hydroxyl ions in a CEM gel phase is much lower than that in the external solution under the conditions of the low concentration tests. According to the microheterogeneous model [20], the concentration of hydroxyl ions in the central part of the mesopores of the CMX and AMX membranes are approximately the same and equal to the concentration of the external solution. However, the proportion of such a solution in both membranes is relatively small—it is no more than 0.1 [21]. However, it was predicted that the OH concentration in the CEM gel phase was almost equal to that in the external solution and almost the same as that in the AEM, when the concentration in the external solution gets closer to the fixed charge density of the CEM.

The purpose of the experiments in this section was to confirm the difference in the effect of alkali degradation on the chemical structure variation between IEMs containing the quaternary ammonium group and sulfonic acid group, when the concentration of hydroxyl ions in the two membranes was almost similar. Hence, the two membranes were immersed in NaOH solutions with a high concentration (6 M), which was almost equal to the fixed charge density of CMX and AMX.

The AMX and CMX used in this study were produced using the paste method [16,17,22]. In this method, the PVC backing cloth is dipped into a paste, which is a mixture of monomer, PVC binder, and initiator. The paste material was polymerized by heating in order to form the precursor for each membrane. Then, sulfonic acid groups were introduced into CMX, while the quaternary ammonium group was introduced into AMX. To confirm the effect of ion-exchange groups, the respective precursors were also immersed in a 6 M NaOH solution at 40 °C. The immersing time periods were 3 h, 24 h (1 day), and 168 h (1 week).

### 2.4. Chemical Analysis

The FTIR spectrum and XRF intensity of the Ka line of chlorine were measured for the samples subjected to alkali immersing tests. The infrared absorption spectra were measured by the Bruker Optics LUMOS (Bruker Japan, Yokohama, Japan) and microscopic attenuated total reflection (ATR) method. All the samples subjected to FTIR were immersed in a 0.5 M NaCl aqueous solution, so that the counter ions of CMX and AMX could be Na ions and Cl ions, respectively. Vacuum-drying was then conducted at 40 °C or less for over 4 h.

The XRF intensity of the Ka line of chlorine was measured by ZSX Primus II (Rigaku, Tokyo, Japan). All the samples were subjected to vacuum-drying at 40 °C or less for 4 h or more, and then they were immersed in a 0.2 M aqueous solution of NaNO_3_, so that the counter ions of CMX and AMX could be Na ions and NO_3_ ions, respectively. The detailed procedures of the experiments were provided in previous papers [18,19].

### 2.5. Characterization of Membrane Properties

The properties of the membranes immersed in low OH concentration tests were characterized. The samples were immersed in a 0.5 M NaCl aqueous solution, so that the counter ions of CMX and AMX were Na ions and Cl ions, respectively. Mechanical and electrical properties (Young’s modulus, water content, ion-exchange capacity (IEC), electrical resistance and hydroxyl ion rejection) of CMX before and after the low OH concentration alkaline immersion tests were characterized, as described in previous papers [18,19].

The hydroxyl ion rejection of the test piece was measured using a craft-made acrylic plastic cell comprising two chambers with a Pt electrode. A 0.5 M NaOH solution and a 3 M NaOH solution were poured into the anode cell and the cathode cell, respectively. Each solution was stirred with a magnetic stirrer and maintained at 25 °C. A current density of 10 A/dm^2^ was applied between the electrodes for 3600 s. The molar number of hydroxyl ions in the cathode cell before the test (*M_1_*) and after the test (*M_2_*) was determined by titration with a 0.05 M H_2_SO_4_ solution to obtain the hydroxyl ion rejection ($P_{OH}$) according to the following equation:(2)$$P_{OH}=\;\frac{M_{2}}{M_{1}}$$

Similarly, the membrane properties before the alkali immersion test was compared with those after the test by calculating the normalized property ratio ($PR_{x}$) as follows:(3)$$PR_{x}=\;\frac{P_{x}^{after}}{P_{x}^{before}}$$
where $P_{x}^{after}$ and $P_{x}^{before}$ denote the membrane properties (e.g., x = Y: Young’s modulus, W: water content, IEC: ion-exchange capacity, ER: electrical resistance, OH: hydroxyl ion rejection, and CO: co-ion rejection) after and before the alkali immersion test, respectively.

## 3. Results and Discussion

### 3.1. Measurement of Alkali Concentration in CMX and AMX

When the IEMs were immersed in a NaOH solution with a concentration of $C_{NaOH}$, the theoretical value of the hydroxyl ion concentration in the IEMs at an equilibrium state ${\bar{C}}_{OH}$ was calculated from the following equations. For simplicity, we assumed that the standard chemical potential difference and activity coefficient of ions in the solution were equal to those of the membrane [23].
(4)$${\bar{C}}_{\mathrm{OH}}=\;C_{\mathrm{NaOH}}K_{don}$$
where $K_{don}$ is the Donnan equilibrium constant, which is calculated from the electroneutrality condition in the membrane and equation of the Donnan theory [23] as follows:(5)$$K_{don}=\frac{z_{x}C_{x}}{2C_{\mathrm{NaOH}}}+\sqrt{{\left(\frac{z_{x}C_{x}}{2C_{\mathrm{NaOH}}}\right)}^{2}+1}\;\;\;\;\;\;\;\;\;\;\;\;\;\;\;\;\;\;\;\;$$
where $z_{x}$ and $C_{x}$ are the valence and concentration of the membrane-fixed charge ($z_{x}$ is 1 for an AEM and −1 for a CEM).

Hydroxyl ion concentrations in CMX and AMX were estimated using these equations. In the case of CMX, $C_{x}$ was calculated as 6M using IEC and W (listed in Table 1), so that the theoretical curve was calculated by substituting $C_{x}$ = 6 M into Equation (5). The theoretical calculations, as well as the experimental values obtained, are shown in Figure 2. The theoretical curve of CMX indicates that the concentration of hydroxyl ions as the co-ions was much lower than that of the external solution. For example, when CMX was immersed in a 0.01 M NaOH solution, the hydroxyl ion concentration in the CMX membrane was 1.7 × 10^−5^ M, which was almost 1/1000 of the concentration of the external solution. While the concentration in the membrane increases as much as that of the external solution, the concentration in CMX was observed to be 0.16 M, even when it was immersed in a 1 M NaOH solution, which was approximately 1/6 of the external solution concentration. In contrast, when CMX was immersed in a 6 M NaOH solution, the concentration was observed to be 3.7 M, which was approximately 60% of the external solution concentration.

In the experiments involving 0.01 M, 1 M, and 6 M NaOH as the external solution, the OH concentration in CMX was estimated at 3.9 × 10^−5^ M, 0.062 M, and 2.5 M, respectively. Each of these experimental observations almost agreed quantitatively with those from the theoretical curve shown in Figure 2. This indicates that the concentration of the co-ions, such as the hydroxyl ions in a CEM, can be estimated quantitatively by substituting the value of fixed charge density into the equation based on the Donnan theory for a wide range of concentrations. The theoretical calculations, as well as the experiments, indicate that the concentration of the co-ions in the CEM was much lower than that of the external solution, when the CEM was immersed in solutions with low OH concentrations.

In the case of AMX, $C_{x}$ was calculated to be 7 M using IEC and W shown in Table 1. Therefore, the theoretical curve was obtained by substituting $C_{x}$ = 7 M into Equation (5). The theoretical values and experimental results obtained at 25 °C are also shown in Figure 2. In contrast to CMX, the theoretical calculations in the case of immersing AMX in the NaOH solution with low concentrations indicate that the concentration of the hydroxyl ions as the counter ions of AMX were approximately equal to 7 M of fixed charge density, regardless of the external solution. When the concentration of the external solution was higher than the fixed charge density, the counter ion concentration in AMX increased with an increase in the external solution concentration and was approximately equal to the external solution concentration. While some deviations were observed between the experimental values and calculations in the case of immersing the AEM in lower NaOH concentrations, the experiments almost agreed quantitatively with the calculations. One of the reasons for this deviation could be that a complete ion exchange between the counter ion of fixed charge and hydroxyl ions in the solution did not occur, when the test piece was immersed in a solution with a low concentration of hydroxyl ions. The test piece was immersed in the NaOH solution for only 1 h to minimize the effects of chemical degradation due to hydroxyl ions. Another possible reason could be that the dehydrochlorination reaction during the immersion test generated hydrochloric acid. The neutralization reaction between the generated hydrochloric acid and hydroxyl ions in the test pieces consumed the hydroxyl ions. Therefore, the OH concentration inside the test pieces were lower than that obtained from the calculations.

Figure 3 and Figure 4 show photographs of CMX and AMX test pieces after immersion in various solutions of NaOH concentrations, respectively. While no color change was observed in all the CMX test pieces, the color of the AMX test pieces changed from transparent to light yellow, orange, and dark violet with increasing NaOH concentrations. This indicates that PVC dehydrochlorination occurred in the test piece of AMX.

A comparison of the CMX and AMX test pieces is shown in Figure 3 and Figure 4 and supported that the concentration of hydroxyl ions in CMX will vary considerably from that in AMX. The theoretical curve shown in Figure 2 indicates that, when AMX is immersed in the 0.01 M NaOH solution and 1 M NaOH solution, its hydroxyl ion concentration is approximately 4 × 10^5^ times and 40 times higher than that of CMX, respectively. As mentioned earlier, it is evident that the dehydrochlorination of CMX did not occur in the case of low OH concentration immersion tests, as the hydroxyl ion concentration in CMX is much lower than that in AMX under the same conditions.

In contrast, when the external solution concentration is almost equal to the fixed charge density, the hydroxyl ion concentration inside CMX is almost equal to that of AMX. Therefore, if the dehydrochlorination reaction is only dominated by the hydroxyl ion concentration during the immersion tests at higher concentrations, dehydrochlorination, similar to that in AMX, is expected to occur in CMX as well. However, even at a high concentration (6 M), no color change was observed in CMX during the immersion test, while AMX changed to dark violet. This phenomenon indicates that the dehydrochlorination reaction is difficult in CMX under these experimental conditions.

A hypothesis from prior reports stated that the quaternary ammonium groups facilitate the dehydrochlorination reaction of PVC [15]. The aforementioned results, shown in Figure 4, can aid in evaluating the validity of this hypothesis. A systematic alkaline immersion test of CMX was performed to confirm this hypothesis in the next section.

### 3.2. Systematic Alkaline Immersion Test (Low OH Concentration Test)

#### 3.2.1. Color Change

Figure 5a shows the color change of CMX test pieces with respect to various NaOH concentrations and immersion time periods at 40 °C. No color change was observed in the CMX test pieces under all the test conditions. The membrane immersed in water for 168 h looked darker than the pristine, but the CMX membrane had color unevenness like the photo of 0.1 M and 3 h. Thus, we concluded that there was no change.

Figure 5b shows the color change of AMX test pieces arranged from the results in a previous paper [18]. In the case of AMX, the pristine test piece was transparent, and no color change was observed after immersion in alkali-free deionized water. In contrast, when they were immersed in alkali solutions, the color gradually changed to yellow, orange, red, and violet, as the immersion time periods increased, and the test pieces eventually turned dark violet. The higher the NaOH concentration, the shorter the immersing time periods required for the change of color to dark violet.

If the hydroxyl ion concentration inside AMX held a constant value according to the calculation curve obtained using the Donnan theory and was almost independent of the external solution concentration, the color change of the test pieces did not depend on the alkali concentration of immersing solutions. However, the colors of the test pieces immersed in a 0.01 M NaOH solution for 3 h, 24 h, and 168 h were yellow, orange, and red, respectively, and that immersed in a 1 M NaOH solution was dark violet. The OH concentration inside AMX decreased with a decline in the external concentration, as shown in Figure 2, which implies that the color change observations were consistent with the experimental results.

#### 3.2.2. Chemical Analysis

##### ATR-FTIR Measurement

As mentioned earlier, a clear difference was observed in the visual color changes of the CMX and AMX test pieces during the alkali immersion tests. In this section, we have discussed the reasons for these color changes to investigate the chemical structure change of the membranes before and after the immersion tests using ATR-FTIR by following the procedure discussed in prior literature [18,19].

Figure 6a shows the comparison of ATR-FTIR spectra of CMX test pieces immersed in 0.01 M, 0.1 M, and 1 M NaOH solutions for 168 h. A spectrum before immersion is also shown here as pristine for comparison purposes. Before the ATR-FTIR measurement, the counter ions of the fixed charge were changed to Na^+^ ions by immersing the test pieces in a 0.5 M NaCl solution for more than 24 h.

For the quantitative discussion, we normalized the obtained spectra using the peak that originated from sulfonic acid group at 1190 cm^−1^, because an investigation of the IEC after and before the immersion tests clearly suggests that the decomposition of the sulfonic acid group has not occurred after the immersion, as shown in Appendix A. The other peaks were also attributed according to the previous literature [15].

In the case of the CMX immersed in lower NaOH concentrations, as expected, had peaks with respect to PVC, which were denoted by the bands at 639 and 690 cm^−1^, corresponding to ν(C–Cl) stretching, those at 1254 cm^−1^ corresponding to δ(CH_2_) wagging, when the adjacent C atom included a chlorine atom, and those at 1425 cm^−1^ due to methylene scissor deformation from PVC, which did not show any change. Furthermore, the broadband intensity observed from 1500–1650 cm^−1^ with respect to the ν(C=C) vibration was not different from that of the pristine test piece. Thus, these spectra clearly indicated that no chemical reactions occurred when CMX was immersed in alkali solutions under 1 M NaOH for 168 h. These results agree with the visual observation results that indicate no color changes in CMX test pieces immersed under such conditions.

Figure 6b shows the comparison of ATR-FTIR spectra of the AMX test pieces immersed in 0.01 M, 0.1 M, and 1 M NaOH solution for 168 h with that of the pristine test piece. In the case of AMX, the counter ions of the fixed charge groups were changed to Cl^−^ ions before the ATR-FTIR measurement. As the aromatic rings of chloromethylstyrene (CMS) and divinyl-benzene (DVB) matrix in AMX were not decomposed after the alkali immersion tests [15], the spectra were normalized using the peak at 1489 cm^−1^, which was attributed to the aromatic ring breathing mode of CMS-DVB, followed by data analysis. In the case of AMX immersed in lower NaOH concentrations, the peaks originating from PVC at 639 cm^−1^, 690 cm^−1^, 1254 cm^−1^, and 1425 cm^−1^ apparently declined after the alkali immersion and gradually declined with an increase in the NaOH concentration. In addition, the intensity of the broadband at 1500–1650 cm^−1^ also increased. Moreover the broadband of 1650–1800 cm^−1^ increased significantly. The broad band of 1600–1800 cm^−1^ can be assigned to an absorbance of conjugated C=C of the resulting polyenes [24]. Thus, these changes in the spectra clearly indicate that PVC dehydrochlorination and subsequent polyene formation have occurred, when AMX was immersed in alkaline solutions even with a lower NaOH concentration (0.01 M). Therefore, these chemical structure changes cause the visual color change of AMX to be more drastic than that of CMX. In contrast, it is worth noting that the intensity of the peak at 1720 cm^−1^, attributed to the (C=O) vibration, also increased, indicating the generation of a carbonyl group through PVC degradation.

##### XRF

Figure 7a,b show the effects of alkali concentration on normalized chlorine intensity using the XRF measurements of CMX and AMX test pieces, each immersed for 168 h. In the case of CMX, no change in the normalized chlorine intensity was observed. In contrast, in the case of AMX, the normalized chlorine intensity decreases, as the NaOH concentration increased. The decrease in the chlorine intensity implies that the PVC dehydrochlorination increased with increasing NaOH concentrations. In particular, approximately 10% of PVC in AMX was lost when the test piece was immersed in 1 M NaOH for 168 h at 40 °C.

#### 3.2.3. Characterization of Membrane Properties

Figure 8 shows the normalized properties of CMX as a function of NaOH concentration and immersion time periods. For comparison, the data corresponding to AMX immersed in 1 M NaOH [18] were also included in these figures.

In the case of AMX, a decrease in the normalized Young’s modulus, an increase in the normalized water content, a decrease in the normalized electrical resistance and co-ion rejection with increasing immersion time periods were observed. In the case of AMX, hydroxyl ions caused the dehydrochlorination of PVC in the membrane, resulting in the formation of polyenes. This polyene formation caused a decrease in the Young’s modulus and an increase in water content. An increase in water content implied an increase in the volume of water channels in the AEM, resulting in a decline in the electrical resistance and co-ion selectivity [18].

In the case of CMX, no change in the normalized Young’s modulus, water content, electrical resistance and co-ion rejection was observed. This is because no changes were observed in the chemical structure of CMX during the low concentration immersing tests, as established by the ATR-FTIR spectra and XRF data.

As CMX showed no changes in its characteristics, it was not necessary to propose a nondestructive method for estimating the electrical characteristics, as was the case for AMX [18,19].

### 3.3. Systematic Alkaline Immersion Test (High OH Concentration Test)

#### 3.3.1. Color Change

Figure 9 shows the color change of the test pieces of CMX, AMX, CMX precursor, and AMX precursor immersed in a 6 M NaOH solution for up to 168 h at 40 °C. When the CMX test piece was immersed in a 6 M NaOH solution, the color change was observed in 24 h. This indicated that even though the CMX was a CEM, it exhibited a color change after immersion in alkali solutions of high OH concentration for long time periods. In contrast, when the AMX test piece was immersed in a 6 M NaOH solution, it immediately turned dark violet, within 3 h, and the color further deepened as the immersion time periods increased. While AMX showed a drastic color change from transparent to dark violet, and CMX exhibited a color change from yellow to orange, no change was observed in the appearance of both CMX precursor and AMX precursor. The CMX precursor contained a styrene and di-vinyl-benzene matrix, while the AMX precursor comprised a chloromethylstyrene and di-vinyl benzene matrix. The former was provided by sulfonic acid groups by the sulfonation reaction, while the latter was provided quaternary ammonium groups by the quaternization reaction with trimethylamine to produce a CEM and an AEM, respectively. The water content of both the precursors was almost zero, because the matrix with no ion-exchange groups was hydrophobic. Hence, the OH ions could not permeate inside the matrix of the two precursors, thereby preventing the PVC dehydrochlorination from occurring even during the severe alkali immersion test conducted with a high OH concentration for longer time periods.

#### 3.3.2. Chemical Analysis

##### ATR-FTIR Measurement

Figure 10a shows the comparison of ATR-FTIR spectrum of the CMX test piece immersed under severe conditions (6 M NaOH solution for 168 h) with that of the pristine test piece. The obtained spectrum was also normalized as there was no decomposition of sulfonic acid group even in the 6 M NaOH case, as shown in Appendix A. In the case of higher NaOH concentrations, the band belonging to the (C–Cl) stretching band of 639 cm^−1^ had declined marginally, and the band at 1600–1700 cm^−1^, which originated from the conjugated (C=C) formation, increased marginally, as compared to those of the pristine test piece. Therefore, these chemical structures changed from the proposed PVC dehydrochlorination and subsequent polyene formation, which had a minimal extent under severe immersion conditions. Therefore, the minimal color change of CMX under severe conditions, shown in Figure 6, was possibly due to polyene formation through PVC dehydrochlorination.

Figure 10b shows the comparison of ATR-FTIR spectrum of the AMX test piece immersed in severe conditions (6 M NaOH solution for 168 h) with that of the pristine test piece. The bands at 639, 690, 1254, and 1425 cm^−1^, corresponding to PVC, significantly declined. The broad band from 1500–1650 cm^−1^ corresponding to (C=C) vibration, and 1650–1800 cm^−1^ corresponding to the conjugated (C=C) vibration significantly increased, compared to those in the case of lower NaOH concentration. Therefore, although the color changes of the AMXs immersed in both higher and lower NaOH concentrations for 168 h appeared to be saturated (the colors in both these cases are dark violet), the PVC degradation and subsequent polyene formation further progressed with increasing NaOH concentrations. This result also agrees with our previous result, which indicates that the membrane performance of AMX declines further with increasing NaOH concentrations, even after the complete change of color to dark violet [18].

In addition, the intensity of the peak at 1720 cm^−1^, attributed to the (C=O) oscillation, also increased significantly.

##### XRF Measurement

Figure 11a,b show the effects of alkali concentration on normalized chlorine intensity using the XRF measurements of CMX and AMX test pieces, each immersed in 6 M NaOH for 168 h. In the case of CMX, no change in the normalized chlorine intensity was observed. This will probably be because XRF could not detect the minimal change in the chlorine content. In contrast, in the case of AMX, the normalized chlorine intensity decreases, as the NaOH concentration increased. Approximately 17% of PVC in AMX was lost.

The normalized chlorine intensities of the CMX precursor and AMX precursor are shown in Figure 11c,d, respectively. It was observed from these figures that the intensity values did not change at all before or after the immersion tests.

### 3.4. Discussion

The results of the chemical analysis of CMX and AMX immersed in alkali solutions are summarized as follows.

#### AMX:

In the ATR-FTIR measurement, the decrease in the peak derived from PVC degradation and the increase in the peak derived from polyene formation were observed. These changes in the peaks increased, as the NaOH concentration increased. In the XRF measurement, the chlorine intensity decreased, as the NaOH concentration increased. These results indicated that the dehydrochlorination of PVC occurred in AMX.

The equation of nucleophilic attack of hydroxide ions is expressed in Scheme 1, called the “E2 elimination reaction”. Hydroxyl ion pull out βhydrogen from PVC, at the same time the chloride ion is eliminated and forms a double bond [22].

In the ATR-FTIR spectra of AMX, the increase in the broadband intensity at 1700–1800 cm^−1^ indicated the generation of carbonyl groups during the alkali immersion test. The formation of the carbonyl band at 1779 cm^−1^ was observed due to sodium hypochlorite attack [14]. However, to the author’s knowledge, carbonyl formation in PVC degradation because of alkali attack has not been reported in the previous literature [11,12,14,15]. Therefore, the mechanism of carbonyl formation should be investigated in the future.

#### CMX:

In the ATR-FTIR measurement, no change was observed in the peak derived from PVC during the immersing tests with up to 1 M NaOH, and no change was observed in the chloride intensity values obtained using the XRF measurement. However, when CMX was immersed under severe conditions (6 M NaOH solution for 168 h), a color change was observed, and the peak indicating the formation of polyene was obtained, but no decrease in chlorine intensity was confirmed by XRF. This will probably be because XRF could not detect the minimal change in the chlorine content. As shown in Appendix A, no change was observed in the IEC of CMX. These observations imply that not much change was observed in the chemical structure of CMX due to alkaline attack.

The dehydrochlorination reaction of PVC in CMX only occurred under severe alkali immersing conditions, such as high NaOH concentrations and long immersing time periods. The dehydrochlorination reaction of PVC scarcely occurred in CMX and cannot be explained by only the low OH concentration inside CMX due to the Donnan equilibrium.

Figure 12 shows a schematic diagram of the distribution of ions inside IEMs in the alkali immersion tests. In the case of the AEMs shown in Figure 12a, quaternary ammonium groups are the positively-charged ones that are fixed with the membrane matrix containing PVC. When the external alkali concentration is almost equal to the fixed charge density, the concentration of the counter-ions (hydroxide ions) in the water phase inside the membrane will be almost equal to that of the cations as the co-ions of the fixed charged groups. However, there will be more hydroxide ions than the cations around the positively-charged groups fixed with the membrane matrix. Hence, the hydroxyl ions can easily attack PVC chains of the membrane matrix. However, in the case of CEM, sulfonic acid groups such as the negatively-charged ones are fixed with membrane matrix, as shown in Figure 12b. Even under the same conditions as the AEM case, there will be more cations than the hydroxide ions around the negatively-charged groups fixed with the membrane matrix. Hence, it will be more difficult for hydroxyl ions to attack PVC chains of the membrane matrix than in the case of AEM. This is the main reason why the dehydrochlorination reaction does not easily occur in CMX.

Vasquez et al. reported that the quaternary ammonium salt in the AEM membrane acts as a catalyst, and dehydrochlorination of PVC with hydroxide ions results in the formation of a polyene sequence [15]. Our immersion tests in severe conditions (6 M NaOH solution for 168 h) revealed that CMX, without any quaternary ammonium groups, changed its color and ATR-FTIR spectrum of CMX indicates that PVC dehydrochlorination and subsequent polyene formation occurred. Therefore, not only AEMs, but also CEMs with a PVC matrix, will be deteriorated by alkali, depending on the conditions. High concentrated hydroxide ions around the positively-charged groups fixed with the matrix will facilitate the alkali degradation of IEMs. One of the ways to avoid alkali degradation of IEMs is to prepare IEMs without PVC materials.

In the future, we will conduct CMX alkali immersion tests at high temperatures and high alkali concentrations to confirm the above hypothesis.

## 4. Conclusions

In this research, to understand the deterioration mechanism of IEMs containing PVC as the backing and binder, systematic experiments were conducted using CMX, which is a commercially available CEM containing PVC in the backing and binder, under various alkali concentrations and immersing time periods at 40 °C. We investigated the changes in the chemical structure and membrane properties of CMX before and after the immersion tests to compare the results with those of a commercial AEM (AMX).

In these experiments, no color change and no membrane properties change was observed up to a concentration of 1 M NaOH in the case of CMX. However, when immersed in 6 M NaOH, which is close to its fixed charge density, for a week, the yellowish membrane turned slightly orange. An analysis of this discolored test piece using ATR-FTIR suggested minimal polyene formation. However, no loss of chlorine was observed in the XRF analysis. These results were compared with those of AMX, as reported in prior papers [17,18]. In the case of AMX, the transparent membrane turned deep violet, and the color change rate increased with an increase in the alkali concentration. The ATR-FTIR analysis indicated that polyene was formed, while the XRF analysis indicated that chlorine was reduced, and PVC was dehydrochlorinated.

ATR-FTIR spectra of the AMX indicated that polyene, as well as the carbonyl groups, were formed through the PVC dehydrochlorination, although the formation mechanism is still unclear. A detailed analysis of PVC dehydrochlorination in an AEM containing PVC is required.

Based on the aforementioned discussion, we propose the mechanism of the differences in alkali degradation in the case of CMX and AMX as follows:

According to the Donnan equilibrium theory, even if CMX and AMX are immersed in NaOH solutions of similar concentrations, which is less than the fixed charge concentration, the hydroxide ion concentration inside CMX is significantly lower than that inside the AMX. This low OH concentration in CMX during the alkali immersion test is a factor of difference with respect to degradation.

The dehydrochlorination reaction of PVC in CMX only occurs under severe alkali immersing conditions: high NaOH concentrations and long immersing time periods. In such scenarios, the OH concentration inside CMX is almost equal to that inside AMX. From these results, it is clear that the dehydrochlorination reaction of PVC scarcely occurs in CMX, not only because of the low OH concentration inside CMX due to the Donnan equilibrium, but also because the negatively charged sulfonic acid group repels the negatively charged hydroxyl ion so that the hydroxyl ion cannot access the PVC that makes up the IPN. This is the main reason why the dehydrochlorination reaction does not easily occur in CMX.

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
