# Peer review of "Alkali Attack on Cation-Exchange Membranes with Polyvinyl Chloride Backing and Binder: Comparison with Anion-Exchange Membranes"

_membranes, 2020, doi:10.3390/membranes10090228_

Round 1

Reviewer 1 Report

The reviewed article continues a series of works that study the effect of aggressive media on the structure and characteristics of ion-exchange membranes. This information is very useful for researchers, industrialists and membrane manufacturers.

The authors describe and discuss the experimental results in detail. The article can be published in the journal Membranes after a minor revision.

My comments and suggestions are as follows.

Lines 93-106 and figure 2 experimental data

The method used by the authors for estimating the concentration of OH ions is very approximate. It gives underestimated values of the concentration of hydroxyl ions in the membrane. The authors could obtain more accurate information using potentiometric titration of a cut portion of the test sample with HCl solution.

Lines 124-126

The phrase contradicts the previous phrase (lines 123-124). The authors probably wanted to say the following: “Thanks to Donnan exclusion of the OH- co-ions, the gel phase of the CMX membrane contains significantly less hydroxyl ions than the gel phase of the AMX membrane. On the other hand, according to the microheterogeneous model [Zabolotsky, V.I., Nikonenko, V.V.

Effect of structural membrane inhomogeneity on transport properties

(1993) Journal of Membrane Science, 79 (2-3), pp. 181-198. DOI: 10.1016 / 0376-7388 (93) 85115-D], the concentration of OH ions in the central part of the mesopores of the CMX and AMX membranes are approximately the same and equal to the concentration of the external solution. However, the proportion of such a solution in both membranes is relatively small; it is no more than 0.1 [Sarapulova, V., et al.  Transport characteristics of fujifilm ion-exchange membranes as compared to homogeneous membranes АМХ and СМХ and to heterogeneous membranes MK-40 and MA-41 (2019) Membranes, 9 (7), art. № 84, . DOI: 10.3390/membranes9070084]»

Fig.5 and 9. I recommend authors to submit color photographs of CMX and AMX membranes. In this case, their color changes will become more pronounced.

Fig.6а,б and 10a,b. I suggest that the authors designate the spectra obtained for different alkali concentrations with different colors, because now these figures are poorly readable.

Figures (b) should be located below figures (a). There should be a single figure caption for each figure.

Fig 6.: “ATR-FTIR spectrum for CMX (a) and AMX (b) membranes before and after soaking in solutions with different NaOH concentrations”

Fig.10: “ATR-FTIR spectrum for CMX (a) and AMX (b) membranes before and after soaking in severe conditions”

On the one hand, the authors write:” …the broadband intensity observed in the range 1500-1650 cm-1 with respect to the v(C = C) oscilations” (line 299). On the other hand, they write:  “…the broad band from 1500-1650 cm-1 corresponding to polyene formation significantly increace…”(lines 399 -400).

Please clarify and give a reference to the literature that allows to make a conclusion about polyenes.

Also, I recommend the author to read the article [Sata, T., Tsujimoto, M., Yamaguchi, T., Matsusaki, K. Change of anion exchange membranes in an aqueous sodium hydroxide solution at high temperature (1996) Journal of Membrane Science, 112 (2), pp. 161-170. DOI: 10.1016/0376-7388(95)00292-8], which presents the results of the behavior of similar membranes with PVC in an alkaline medium.

Author Response

Thank you very much for providing helpful comments which leads to improve our manuscript (Membranes-927114). The answers for two reviewer’s comments are shown as followings. In the revised manuscript, the revision parts have written in red characters and indicate yellow highlight.

The reviewed article continues a series of works that study the effect of aggressive media on the structure and characteristics of ion-exchange membranes. This information is very useful for researchers, industrialists and membrane manufacturers.

The authors describe and discuss the experimental results in detail. The article can be published in the journal Membranes after a minor revision.

Thank you very much for the favorable evaluation.

My comments and suggestions are as follows.

  1. Lines 93-106 and figure 2 experimental data

The method used by the authors for estimating the concentration of OH ions is very approximate. It gives underestimated values of the concentration of hydroxyl ions in the membrane. The authors could obtain more accurate information using potentiometric titration of a cut portion of the test sample with HCl solution.

(answer)

Thank you for your helpful comment. Unfortunately, we do not currently have any potentiometric titration system. But, we would like to set up and use the system before our next study.

  1. Lines 124-126

The phrase contradicts the previous phrase (lines 123-124). The authors probably wanted to say the following: “Thanks to Donnan exclusion of the OH- co-ions, the gel phase of the CMX membrane contains significantly less hydroxyl ions than the gel phase of the AMX membrane. On the other hand, according to the microheterogeneous model [Zabolotsky, V.I., Nikonenko, V.V. Effect of structural membrane inhomogeneity on transport properties (1993) Journal of Membrane Science, 79 (2-3), pp. 181-198. DOI: 10.1016 / 0376-7388 (93) 85115-D], the concentration of OH ions in the central part of the mesopores of the CMX and AMX membranes are approximately the same and equal to the concentration of the external solution. However, the proportion of such a solution in both membranes is relatively small; it is no more than 0.1 [Sarapulova, V., et al.  Transport characteristics of fujifilm ion-exchange membranes as compared to homogeneous membranes АМХ and СМХ and to heterogeneous membranes MK-40 and MA-41 (2019) Membranes, 9 (7), art. № 84, . DOI: 10.3390/membranes9070084]»

(answer)

Thank you for your helpful comment.

In line with your comment, we revised the revised manuscript as follows:

Lines 124-132 in the revised manuscript.

“According to the Donnan exclusion, the concentration of hydroxyl ions in a CEM gel phase is much lower than that in the external solution under the conditions of the low concentration tests. According to the microheterogeneous model [22], the concentration of hydroxyl ions in the central part of the mesopores of the CMX and AMX membranes are approximately the same and equal to the concentration of the external solution. However, the proportion of such a solution in both membranes is relatively small; it is no more than 0.1 [23]. On the other hand, it is predicted that the OH concentration in the CEM gel phase is almost equal to that in the external solution and almost the same as that in the AEM, when the concentration in the external solution gets closer to the fixed charge density of the CEM. “

  1. 5 and 9.
    I recommend authors to submit color photographs of CMX and AMX membranes. In this case, their color changes will become more pronounced.

(answer)

Thank you for your comment. In line with your comment, when submitting the revised manuscript, we will send color photographs of all the membranes with the manuscript.

  1. 6а,b and 10a,b. I suggest that the authors designate the spectra obtained for different alkali concentrations with different colors, because now these figures are poorly readable.

(answer)

The IR system automatically draws the spectrum; hence, we can not change the colors. But, in response to your helpful comment, we will send more readable figures (with high resolution) for the spectrum, when submitting the revised manuscript.

  1. Figures (b) should be located below figures (a). There should be a single figure caption for each figure.

Fig 6.: “ATR-FTIR spectrum for CMX (a) and AMX (b) membranes before and after soaking in solutions with different NaOH concentrations”

Fig.10: “ATR-FTIR spectrum for CMX (a) and AMX (b) membranes before and after soaking in severe conditions”

(answer)

Lines 335-336 and 423-424 in the revised manuscript.

Thank you for your comment. In line with your comment, we have revised the revised manuscript.

  1. On the one hand, the authors write:” …the broadband intensity observed in the range 1500-1650 cm-1 with respect to the v(C = C) oscilations” (line 299). On the other hand, they write: “…the broad band from 1500-1650 cm-1 corresponding to polyene formation significantly increace…”(lines 399 -400).
    Please clarify and give a reference to the literature that allows to make a conclusion about polyenes.

(answer)

Thank you for your helpful comment. In response to the comment, we have revised the manuscript as:

Lines 323-330 in the revised manuscript.

“Moreover the broadband of 1650-1800 cm-1 increased significantly. The broad band of 1600-1,800 cm-1 can be assigned to an absorbance of conjugated C=C of the resulting polyenes [24]. Thus, these changes in the spectra clearly indicate that PVC dehydrochlorination and subsequent polyene formation have occurred, when AMX was immersed in alkaline solutions even with a lower NaOH concentration (0.01 M). Therefore, these chemical structure changes cause the visual color change of AMX to be more drastic than that of CMX. In contrast, it is worth noting that the intensity of the peak at 1720 cm-1 attributing to the (C = O) vibration also increased, indicating the generation of carbonyl group through PVC degradation.”

  1. Also, I recommend the author to read the article [Sata, T., Tsujimoto, M., Yamaguchi, T., Matsusaki, K. Change of anion exchange membranes in an aqueous sodium hydroxide solution at high temperature (1996) Journal of Membrane Science, 112 (2), pp. 161-170. DOI: 10.1016/0376-7388(95)00292-8], which presents the results of the behavior of similar membranes with PVC in an alkaline medium.

(answer)

Thank you for your comment. Actually, we have also read Sata's paper and cited it in the first and second papers for the degradation of AEMs because Dr. Sata studied on the alkaline degradation of PVC-containing AEMs. We do not cite this paper in this manuscript because our main target in the manuscript is degradation of cation-exchange membranes. Moreover, the experiments by Sata were used a highly concentrated alkali of 3M or more, and it is different from the concentration in CIP conditions (about 0.1M) performed at the site. In our research series, we conducted a systematic experiment at lower concentrations than the study by Dr. Sata.

Reviewer 2 Report

This study systematic investigated the alkali attack on both AEM and CEM with PVC as the backing and binder of that ion exchange membranes. The authors found the CEM with PVC was deteriorated by strong alkali under severe conditions (6 M for 168 h at 40°C) and the CEM was not significant affected by a relative low concentration alkali. This manuscript is well organized and also provides some basic information for the alkali tolerance of the commercial ion exchange membranes. Therefore, I suggest for the publication of the manuscript after minor revision. 1. This is little information in figure 1. Delete this figure or providing it as supporting information is suggested. 2. In line 69, please explain the meaning of SS. 3. In figure 5, it seems that color is changed for CMX that is immersed in water for 168h compared with that of the pristine membrane. 4. In section 3.3, the authors compared the properties of the AEM and CEM with the precursors of that membranes. But the membrane precursors were not functionalized to introduce the functional groups. This kind of comparison is not convinced. Why not compared the membranes with that without PVC as backing? 5. All the membranes were just immersed in an alkaline aqueous solution, there is quite different from the practical application of the membranes. In practical applications, the membranes were operated at the presence of current field, both the current and temperature will affect the main properties of the membranes. Are the conclusions in the study also suitable for practical applications of the ion exchange membranes? 6. There are some typo errors in the superscript and subscript, such in Line 309, Line 312.

Author Response

Reviewer 2

Thank you very much for providing helpful comments which leads to improve our manuscript (Membranes-927114). The answers for two reviewer’s comments are shown as followings. In the revised manuscript, the revision parts have written in red characters and indicate yellow highlight.

This study systematic investigated the alkali attack on both AEM and CEM with PVC as the backing and binder of that ion exchange membranes. The authors found the CEM with PVC was deteriorated by strong alkali under severe conditions (6 M for 168 h at 40°C) and the CEM was not significant affected by a relative low concentration alkali. This manuscript is well organized and also provides some basic information for the alkali tolerance of the commercial ion exchange membranes. Therefore, I suggest for the publication of the manuscript after minor revision.

Thank you very much for the favorable evaluation.

  1. This is little information in figure 1. Delete this figure or providing it as supporting information is suggested.

(answer)

Thank you for your comment. But, we think the photo is indicating what is occurred in the cite where ion-exchange membranes are used by the alkali cleaning processes.  And the phenomenon was an opportunity to start this research. Hence, we think that the posting of this figure is valid for the paper.

  1. In line 69, please explain the meaning of SS.

(answer)

Thank you for your comment. In response to the comment, we have added the sentence: “Stress-Strain curve obtained in tensile test” in line 70 in the revised manuscript.

In figure 5, it seems that color is changed for CMX that is immersed in water for 168h compared with that of the pristine membrane.

(answer)

Thank you for your helpful comment. As you mentioned, the membrane immersed in water for 168 h looks darker than the pristine. CMX membrane has color unevenness like the photo of 0.1 M and 3 h. Hence, we have changed the text as “The membrane immersed in water for 168 h looks darker than the pristine, but CMX membrane has color unevenness like the photo of 0.1 M and 3 h. So we concluded there is no change.” in line 264-266 in the revised manuscript.

  1. In section 3.3, the authors compared the properties of the AEM and CEM with the precursors of that membranes. But the membrane precursors were not functionalized to introduce the functional groups. This kind of comparison is not convinced. Why not compared the membranes with that without PVC as backing?

(answer)

Thank you for your helpful comment. As you pointed out, now, we are comparing the alkali degradation of membranes having PVC with those without PVC. We are able to show the paper in near future.

  1. All the membranes were just immersed in an alkaline aqueous solution, there is quite different from the practical application of the membranes. In practical applications, the membranes were operated at the presence of current field, both the current and temperature will affect the main properties of the membranes. Are the conclusions in the study also suitable for practical applications of the ion exchange membranes?

(answer)

As you pointed out, the current and temperature during operation affect the deterioration of the membrane.

In the actual field, CIP is used to wash the equipment with alkali without applying electric current between operations. We think that repeated alkaline cleaning in the non-energized state also contributes to the deterioration of the IEMs. The effect of applied current and operating temperature on the deterioration of the membranes is our future research topic.

  1. There are some typo errors in the superscript and subscript, such in Line 309, Line 312.

(answer)
Thank you for your comment. In response to the comment, we have revised the typo errors.
